# Spatial-Temporal Heterogeneity of Ecosystem Service Value Driven by Nature-Human Activity-Policy in a Representative Fragile Karst Trough Valley, SW China

Cheng Zeng [†], Gaoning Zhang [†], Tianyang Li [iD], Binghui He *[iD] and Dengyu Zhang

College of Resources and Environment, Southwest University, Chongqing 400715, China;
zengcheng@email.swu.edu.cn (C.Z.); zhanggaoning@email.swu.edu.cn (G.Z.); tyli53@swu.edu.cn (T.L.);
zdy942810302@email.swu.edu.cn (D.Z.)
* Correspondence: hebinghui@swu.edu.cn or hebinghuiswu@163.com; Tel.: +86-23-68251249
† These authors contributed equally to this work.

**Abstract:** Most studies on the ecosystem service value (ESV) only focus on spatial/temporal heterogeneity or single driving effects, but little is known about the combined effects of nature-human activity-policy on ESV in the fragile karst areas. This study aimed to investigate the spatial-temporal heterogeneity of ESV between 1990 and 2020 in a representative karst trough valley in SW China. The dynamic degree of land use, the land-use transfer matrix, sensitive analyses, Geo-Detector, and Hot- and cold-spots analyses were used to determine the interactions between ESV and the natural, human activity, and policy-driving factors. The results showed that from 1990 to 2020, forestland accounted for more than 62.32% of the total area, and construction land increased continuously as a result of urbanization. The conversion from forestland to cropland dominated the land-use transfer. The ESV decreased sharply between 1990 and 2000 and slowly increased from 2000 to 2020, causing a total reduction of 562.91 million yuan in ESV, due largely to the occupation of forestland and grassland by cropland. Soil type (8.00%) had the largest explanation rate for the spatial-temporal heterogeneity of ESV, followed by population density (7.71%) and altitude (6.34%). Policy factors not only markedly influenced the ecosystem structure and function and tradeoff and synergy but also regulated their interactions with other driving factors. Our results have great significance for understanding the interaction effect of natural sources and human activities on changes in ESV in karst areas.

**Keywords:** ecosystem service value; influencing factor; spatial-temporal evolution; Karst





## 1. Introduction

Ecosystem service values (ESVs) are the services and products that humans obtain directly or indirectly from the structure, functions, and processes of ecosystems [1,2]. ESV plays a crucial role in realizing the transformation of ecological assets [3]. In the context of global environmental change and urbanization, human activities and land use have intensified the destruction of the ecological environment and led to the degradation of ecosystem service functions and structures [4], which in turn poses a great threat to the sustainable development of society and economy [5,6].

Previous researchers have studied the spatial and/or temporal changes in ESV in many different ecosystems worldwide. Pham et al. [7] investigated the impacts of land-use transition on ESV to interpret the urbanization pattern effects on the ecosystems in Nha Trang, Vietnam. Mengist et al. [8] studied the spatial-temporal variation of ESV and its relationship with LUCC dynamics in the Kaffa Biosphere Reserve, and found that the decline of ESV resulted from the land-use change that caused ecosystem degradation. Cao et al. [9] assessed the relationships between ESV and regional urbanization characteristics as well as economic growth in China between 2000 and 2015. Basu et al. [10] estimated the loss of ESV due to urban expansion using the urban growth prediction model and the

Markov Chain and Multi-Layer Perceptron Neural Network in the Gateway of North-East India. Although great efforts have been made to the changes in ESV at the regional and/or national ecosystems, the changes in ESV in some specific and fragile ecosystems, such as the karst area, remain largely unclear.

The karst ecosystem is one of the most specific and fragile ecosystems in the world due to severe rocky desertification caused by unique geological structures and unreasonable human activities [6,11]. Karst landforms measure approximately 22 million km$^2$ in the world, accounting for approximately 15% of the global land area. China is one of the regions with the largest karst landscape in the world. The total area of exposed karst reaches $1.3 \times 10^6$ km$^2$, accounting for approximately 13.5% of China's total area [12]. The karst mainly distributed in southwest China, and is characterized by a typical ecologically fragile landscape. The concentrated distribution of carbonate rocks and rock desertification results in specific environmental characteristics of shallow soil layers and underlying surface fragments in the karst ecosystem [13]. The intensive human activities in the karst area have degraded ecological carrying capacity, and once it is damaged, serious ecological problems occur [14]. Therefore, the accurate calculation of ESV is of great significance for protecting the normal operation of the ecosystem and preventing regional ecological risks. Given the harshness of karst ecosystems, it is necessary to accurately evaluate the spatial-temporal variations in ESV to ensure ecological security and ecological restoration in karst areas.

To date, few studies have focused on the spatial and/or temporal heterogeneity in ESV in karst areas. Wang et al. [15] estimated the changes in ESV in the karst area of Guizhou province between 2000 and 2020 and quantified the contribution of land-use transitions to the ESV change. Zhang et al. [16] explored the evolution process of ESV by standard deviation ellipse and spatial autocorrelation analysis in the karst-Beibu Gulf of southwest Guangxi. Zhang et al. [17] studied the response of ESV to landscape patterns in typical karst areas of northwest Guangxi between 1985 and 2005. However, the fragility characteristics of the karst ecosystem make it difficult to restore and cause it to be a very prominent environmental problem [18]. The harsh natural environment and unreasonable human activities affect the ecosystem structure and function in karst areas, thus driving the change in ESV at the regional scale [11]. In addition, the implementation of a series of ecological restoration policies and urbanization displayed complex impacts on karst ecosystems [19]. Therefore, when interpreting the spatial-temporal changes in ESV in karst areas, it is necessary to jointly investigate the effects of natural factors, human activities, and policy factors on the spatial-temporal changes in ESV and its driving mechanisms.

In this study, Youyang County, a representative karst trough valley, was identified as the study area, and the dynamic degree (DD) of land use and the land-use transfer matrix were calculated to reveal the process of LUCC pattern change in terms of the land use and land cover change (LUCC) in 1990, 2000, 2010, and 2020. Based on the equivalent factor method and the actual situation of Youyang County, the ESV in different periods were calculated. On the basis of quantifying the spatial-temporal evolution characteristics of ESV, sensitivity analysis, Geo-Detector, and hot- and cold-spot analysis methods were used to comprehensively determine the contributions and driving mechanisms of various factors (i.e., natural factors, human activities, and policy factors) to the changes in ESV. The results can provide a theoretical basis and data support for ecological restoration and management in karst areas worldwide.

## 2. Materials and Methods

### 2.1. Study Region

This study is conducted in the Youyang Tujia and Miao Autonomous County (Youyang County) (108°18′25″–109°19′02″ E, 28°19′28″–29°24′18″ N), which is located southeast of the Chongqing Municipality (Figure 1). Youyang County covers an area of 517,300 ha and is a typical karst trough valley that belongs to the Wuling Mountain. The highest point is within northern Youyang County, with an altitude of 1895 m. Youyang County is located

in the subtropical humid monsoon climate zone. The annual precipitation is concentrated from April to October, accounting for 84.11% of the total annual rainfall, with an average annual precipitation of 1353 mm. The average annual temperature is 11 °C with the coldest temperature of 1 °C in January and the highest of 30 °C in August.

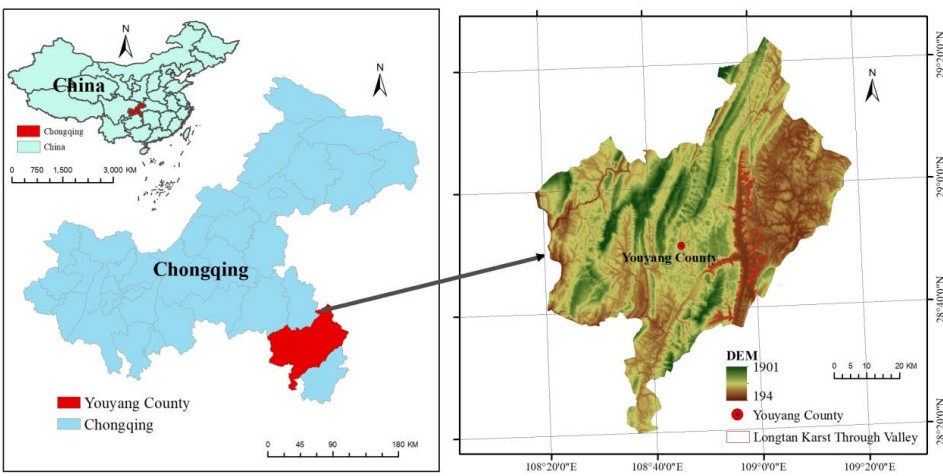

**Figure 1.** Location map of Youyang County.

Between 1990 and 2000, severe droughts and floods sequentially occurred in southern China. At the same time, the Chinese government encouraged economic development, thereby resulting in the expansion of cropland and increases in grain yield and population and the start of urbanization. These changes strongly altered the types and functions of land use. Due to the fragile geological and geomorphic environment of Youyang County, the ecological environment suffered from intensive destruction at a large scale during this period. Between 2000 and 2010, a series of ecological restoration policies were implemented by the government of China. The proportion of the non-agricultural population rose slowly in Youyang County; however, the industrial distribution was dominated by agriculture. Meanwhile, urbanization developed slowly, and the deterioration of the ecological environment began to be controlled. Between 2010 and 2020, the ecological restoration policy implemented over the past decade played a prominent role. With the development and transformation of the rural economy, the construction of main functional zones in Chongqing and the migration of a large number of migrant workers caused significant declines in the proportion of the agricultural population and the transformation of land-use function, thus decreasing the enhancement of living function and ecological function and the agricultural production function.

### 2.2. Data Sources

LUCC data were obtained from the global land cover remote sensing mapping and key technology project GlobeLand30 (http://www.globallandcover.com/, accessed on 1 March 2023), with a spatial resolution of 30 m and an overall accuracy of 83.50%. Based on the LUCC from the resource and environment science data center of the Chinese Academy of Sciences (https://www.resdc.cn/, accessed on 1 March 2023), LUCC data in Youyang County were re-classified into 5 categories: forestland, cropland, grassland, construction land, and water body (Figure 2). The digital elevation model (DEM) was derived from the geospatial data cloud (http://www.gscloud.cn, accessed on 1 March 2023) with a spatial resolution of 30 m. DEM was used to calculate the slope degree, slope aspect, and altitude. The normalized difference vegetation index (NDVI) and net primary productivity (NPP) data are from the resource and environment science data center of the Chinese Academy of Sciences (https://www.resdc.cn/, accessed on 1 March 2023) with a resolution of 500 m. All raster data were resampled into a spatial resolution of 30 m. Agricultural product prices, population density, and other socio-economic data (unit area grain yield

and grain market price) were compiled from the Youyang Statistical Yearbook (1990–2020) and China Price Yearbook (1990–2020). To conduct standardization and accurate statistics of ESV data, ArcGIS 10.2 was used to create a fishing net tool, and Youyang County was divided into 5173 square grid units (1000 m × 1000 m) for the evaluation of the ESV spatial-temporal pattern.

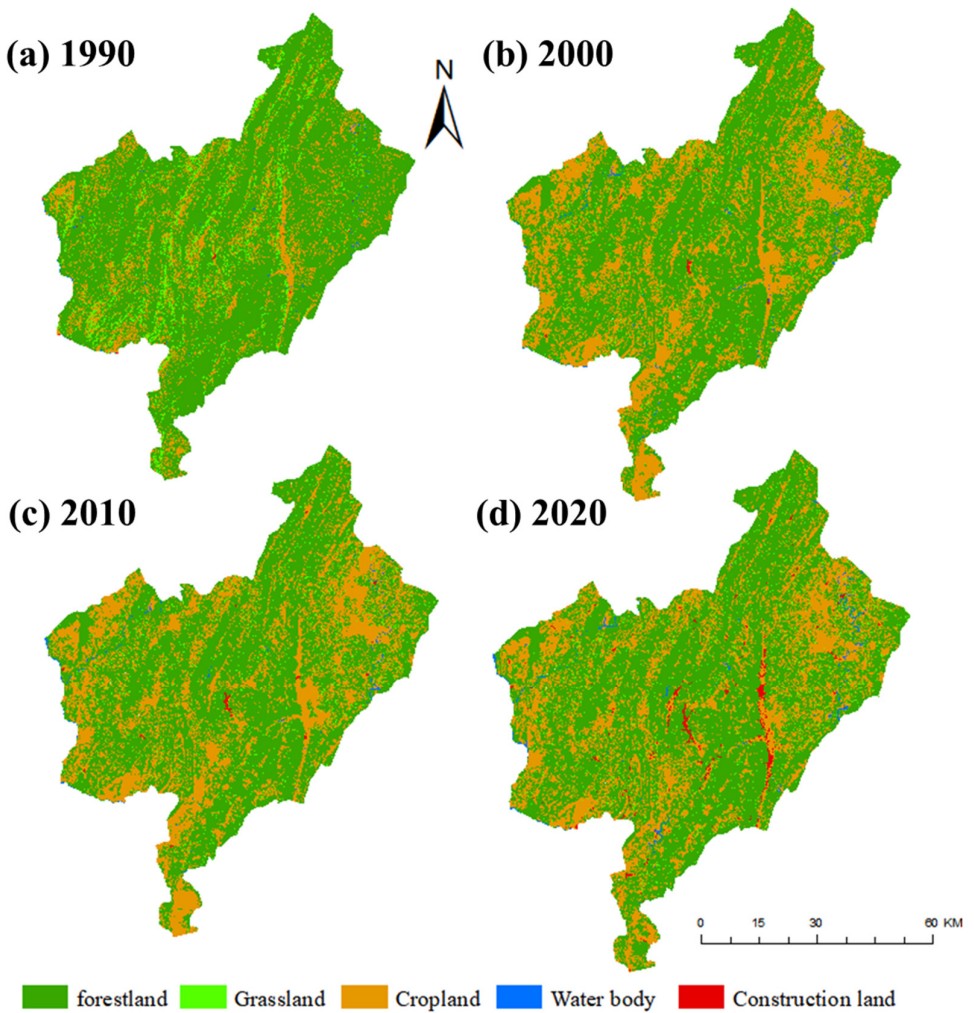

**Figure 2.** Land use and land cover change (LUCC) maps of Youyang County in 1990 (**a**), 2000 (**b**), 2010 (**c**), and 2020 (**d**).

*2.3. Dynamic Change in LUCC*

2.3.1. Dynamic Degree of Land Use

Dynamic degree (DD) of land use presents the intensity of change in land-use type and quantity in Youyang County in a certain period, which plays an important role in predicting the trend of land-use changes and analyzing regional heterogeneity [20]:

$$R_i = \frac{U_b - U_a}{U_a} \times \frac{1}{T} \times 100 \tag{1}$$

where $R_i$ is the dynamic degree of a certain type of land use; $T$ is the change time of a certain land-use type; and $U_a$ and $U_b$ are the areas of a certain land-use type at the beginning and the end, respectively.

### 2.3.2. Land-Use Transfer Matrix

The land-use transfer matrix is used to depict the transfer direction and quantity change among land-use types and can reveal the evolution processes of land-use patterns [21]. The land-use data of Youyang County was fused and intersected in 1990–2000, 2000–2010, and 2010–2020, respectively, using ArcGIS 10.2 software. The land-use transfer matrix was established through the PivotTable of Microsoft Excel LTSC (version 2108) software [22]:

$$A_{xy} = \begin{bmatrix} A_{11} & A_{12} & \cdots & A_{1n} \\ A_{21} & A_{22} & \cdots & A_{2n} \\ \vdots & \vdots & \vdots & \vdots \\ A_{n1} & A_{n2} & \cdots & A_{nn} \end{bmatrix} \tag{2}$$

where $A_{xy}$ is the land area that was $x$ type at the beginning of the study and was converted to the $y$ l type at the end of the study; $n$ is all types of land use.

### 2.4. ESV Accounting

The equivalent coefficients table and the standard equivalent factor were used to evaluate the ESV in Youyang County [1,23]. After comprehensive comparative analysis, the equivalent coefficient of ESV in Youyang County was transformed by 1/7 of the economic value of grain production in the current year [23]. Then, the equivalent coefficient of ESV in Youyang County was modified and listed in Table 1. Finally, the ESV was obtained by the following equation:

$$E_v = 1/7 \sum_{i=1}^{n} \frac{m_i p_i q_i}{M} \quad (i = 1 \dots n) \tag{3}$$

where $E_v$ is the economic value of grain production services provided by farmland ecosystem per unit area (yuan/ha), $i$ is the crop type, $m_i$ is the grain area (ha), $p_i$ is the average unit price of grain crops (yuan/kg), $q_i$ is the grain yield of per unit area (kg/ha), and $M$ is the total area of all crops (ha). The agricultural price in other years has been converted to the constant price in 2020.

$$V = E \times z \tag{4}$$

$E$ refers to the economic value of providing food services for cropland per unit area; $z$ refers to the equivalent coefficient of ESV per unit area according to Xie et al. [23]. The ESV equivalents per unit area of different land types in Youyang County were then determined (Table 1)

$$ESV = \sum_{i=1}^{5} V_i \times S_i \tag{5}$$

where $V_i$ is the equivalent coefficient of $i$th land use in Youyang County; $S_i$ is the area of $i$th land use.

**Table 1.** The equivalent coefficient of ESV in Youyang County.

| Ecosystem Service Type | Ecosystem Service Function | ESV Equivalent Value Coefficient (Million yuan/ha) | | | | |
| --- | --- | --- | --- | --- | --- | --- |
| | | Cropland | Forestland | Grassland | Water Body | Construction Land |
| Provision | Food Production | 865.96 | 197.88 | 188.08 | 626.94 | 0.00 |
| | Raw Material Production | 192.00 | 454.53 | 274.28 | 180.24 | 0.00 |
| | Water Supply | 1038.36 | 235.10 | 152.82 | 6496.63 | 0.00 |
| Regulate | Gas Regulation | 697.47 | 1494.85 | 971.75 | 603.43 | 0.00 |
| | Climate Regulation | 364.41 | 4472.80 | 2566.52 | 1794.60 | 0.00 |
| | Purification of the Environment | 105.80 | 1310.69 | 846.36 | 4349.37 | 0.00 |
| | Water Regulation | 1171.59 | 2927.01 | 1880.81 | 80,122.45 | 0.00 |
| Support | Soil Formation and Retention | 407.51 | 1820.07 | 1183.34 | 728.81 | 0.00 |
| | Maintenance of Nutrient Circulation | 121.47 | 139.10 | 90.12 | 54.86 | 0.00 |
| | Biodiversity | 133.22 | 1657.46 | 1073.63 | 1998.36 | 0.00 |
| Culture | Recreation, Culture, Tourism | 58.78 | 726.85 | 474.12 | 1481.14 | 0.00 |

*2.5. Sensitivity Analysis of ESV*

The Coefficient of Sensitivity (CS) was used to reflect the dependence of ESV changes on the value coefficient (VC) of ecosystem services in a certain landscape [24]. The larger the CS, the more sensitive the ESV was to the VC change in this kind of landscape, indicating a higher contribution to ESV. By adjusting the VC of land use up and down by 50%, the response of ESV to the change in VC was calculated as follows [25]:

$$CS = \left| \frac{(ESV_j - ESV_i) / ESV_i}{(VC_j - VC_i) / VC_i} \right| \tag{6}$$

where $ESV_i$ is the initial ESV and $ESV_j$ is the adjusted ESV. $VC_i$ and $VC_j$ are the VCs of ecosystem services before and after adjustment, respectively.

*2.6. Driving Mechanism of Spatial-Temporal Heterogeneity of ESV*

2.6.1. Geo-Detector

A factor detector and an interactive detector were determined to detect the influence of natural factors and human activities on ESV in the karst trough valley [25]. Among them, the factor detector was employed to determine the spatial heterogeneity of ESV and the explanation degree of the driving factor to the spatial differentiation of ESV. The interaction detector was used to explore whether the combined effects of different factors increase or decrease the explanation degree of ESV [26].

$$q = 1 - \frac{1}{N\sigma^2} \sum_{n=1}^{L} N_h \sigma_h^2 \tag{7}$$

where $q$ is the explanatory degree of the driving factor. It quantifies the spatial-temporal heterogeneity of the dependent variables including NDVI, NPP, altitude, slope degree, slope aspect, lithology, soil type, and population density; $N$ is the number of all units that can be divided into $L$ strata; Stratum h comprised $N_h$ units. Commonly, $q$ falls within the range from 0 to 1, and a larger $q$ presents stronger explanatory power of the corresponding factor.

2.6.2. Hot- and Cold-Spots Analyses

Policies directly drive land-use patterns and change the spatial-temporal distribution of habitats and resources, and thus affecting the structure and function of ecosystems and indirectly driving spatial-temporal changes in ESV [27]. Hot- and cold-spot analyses are widely used to display the spatial distribution of the corresponding study variables for enhancing socio-economic and policy changes [28,29]. The spatial clustering of ESV changes was explored to identify the locations of spatially statistically significant hot-spots (high values) and cold-spots (low values) using ArcGIS 10.2 software in Youyang County from 1990 to 2020. Based on the land-use changes and selected driving factors, we compared the spatial distribution characteristics of hot- and cold-spots of ESV change so as to qualitatively assess the positive or negative regulatory effects of policy factors on ESV during the study period and explore the driving effects of policy factors on spatial-temporal changes in ESV.

$$G_i^* = \sum_{j}^{n} W_{ij}(d) x_i \Big/ \sum_{j}^{n} x_i \tag{8}$$

$$Z\left(G_i^*\right) = \frac{G_i^* - E\left(G_i^*\right)}{\sqrt{Var\left(G_i^*\right)}} \tag{9}$$

where $G_i^*$ is the aggregation index of $i$ of space unit, Z is the significance of the aggregation index, $E\left(G_i^*\right)$ is the mathematical expectation of $(G_i^*)$, Var $(G_i^*)$ is the variance of $(G_i^*)$, and $W_{ij}$ represents the spatial weight value. If the calculation result of $Z\left(G_i^*\right)$ is positively significant, it means that the values around $i$ are relatively high (higher than the mean) and belong to the spatial agglomeration of high values (hot-spot area). In contrast, if the

calculation result of $Z$ $(G_i^*)$ is negatively significant, it denotes that the values around $i$ are relatively low (lower than the mean) and belong to low-value spatial agglomeration (cold-spot area).

## 3. Results

### 3.1. Dynamics of LUCC

#### 3.1.1. Spatial-Temporal Variations of LUCC

Forestland was always the dominant LUCC in Youyang County, accounting for 63.32–76.94% of the total area from 1990 to 2020 and exhibiting a relatively uniform distribution in the ecosystem (Table 2).

**Table 2.** The area of LUCC change in Youyang County from 1990 to 2020.

| Year | Cropland | | Forestland | | Grassland | | Water Body | | Construction Land | |
|------|-----------|-------|-------------|-------|------------|------|------------|------|--------------------|------|
| | ha | % | ha | % | ha | % | ha | % | ha | % |
| 1990 | 85,165.07 | 16.46 | 397,986.95 | 76.94 | 33,216.60 | 6.42 | 694.91 | 0.13 | 236.47 | 0.05 |
| 2000 | 171,908.01 | 33.23 | 327,561.02 | 63.32 | 15,489.88 | 2.99 | 1931.54 | 0.37 | 409.55 | 0.08 |
| 2010 | 171,128.40 | 33.08 | 327,990.84 | 63.40 | 15,553.43 | 3.01 | 1881.94 | 0.36 | 745.39 | 0.14 |
| 2020 | 158,070.09 | 30.56 | 335,327.56 | 64.82 | 15,362.05 | 2.97 | 2743.00 | 0.53 | 5797.30 | 1.12 |

Forestland decreased rapidly by 70,425.93 ha between 1990 and 2000 and slowly increased by 7766.54 ha between 2000 and 2020 (Figure 2). Cropland was the secondary LUCC type explaining 16.46–33.23% of the total area and was mostly distributed in the Longtan karst trough valley and the middle-low mountain areas (altitude below 1000 m) (Table 2). Cropland expanded rapidly by 86,742.94 ha to the surrounding cities and towns between 1990 and 2000 and slowly contracted by 13,837.92 ha between 2000 and 2020 (Figure 2).

Grassland occupied 2.97–6.42% of the total area and was adjacent to the cropland and rivers (Table 2). It decreased rapidly by 17,726.72 ha between 1990 and 2000 and kept at a constant area but dispersed in spatial distribution between 2000 and 2020 (Figure 2). Water bodies accounted for only 0.13–0.53% of the total area (Table 2) and was distributed sparsely with weak spatial variation (Figure 2). Construction land explained 0.05–1.12% of the total area and was mainly distributed in strips in the urban area of Youyang County and the Longtan karst trough valley (Table 2). Construction land displayed an expanding trend from 1990 to 2020 (Figure 2).

#### 3.1.2. Conversion between LUCC

From 1990 to 2020, the DD of construction land reached 225.52% and was much higher than that of other land-use types (Figure 3). The DD of water body (27.92%) and cropland (7.83%) increased, and that of forestland (−1.91%) and grassland (−5.56%) both decreased. Between 1990 and 2000, the DD of water bodies (16.72%) and cropland (9.40%) were relatively higher than that of construction land (6.65%), while forestland (−2.09%) and grassland land (−5.52%) were markedly reduced. Between 2000 and 2010 and 2010 and 2020, the DD of construction land was 8.20% and 67.72%, respectively, and the dynamic degrees of other land-use types changed slightly.

The land-use internal transfer was significantly different between the periods of 2000–2010 and 2010–2020 (Figure 4). From 1990 to 2020, 24.53% of forestland (97,015.17 ha) and 19.28% of grassland (6056.92 ha) were converted to cropland, respectively. Furthermore, 28.70% of cropland (23,544.15 ha) and 78.80% of grassland (24,752.47 ha) were transformed into forestland. Water bodies were derived from forestland (1535.65 ha) and cropland (492.25 ha), accounting for 57.33% and 18.38% of the water bodies in 2020, respectively. Construction land stemmed from cropland (3803.84 ha) and forestland (1574.47 ha), accounting for 66.71% and 27.61% of the construction land in 2020, respectively.

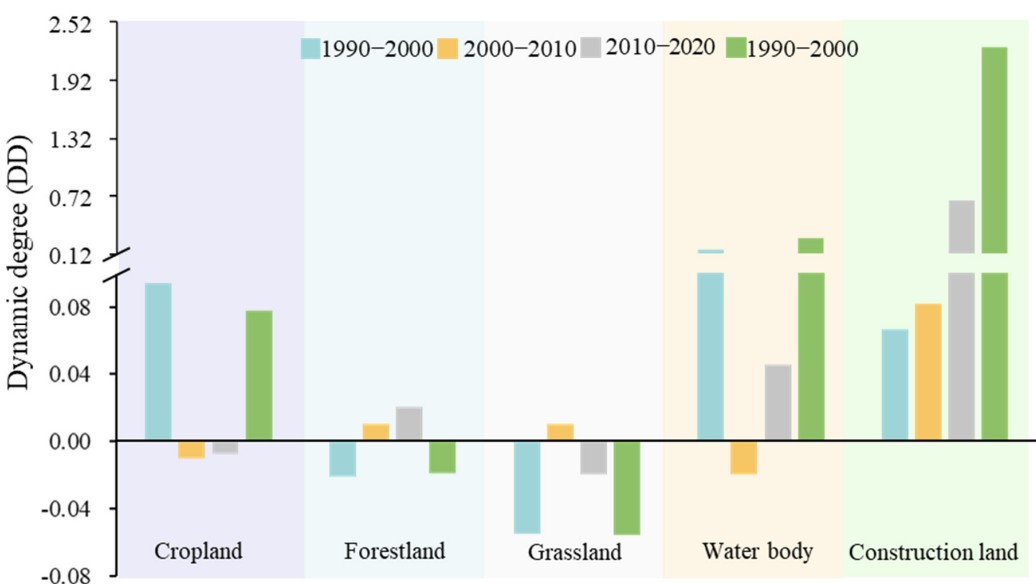

**Figure 3.** Dynamic degrees (DD) of land use from 1990 to 2020.

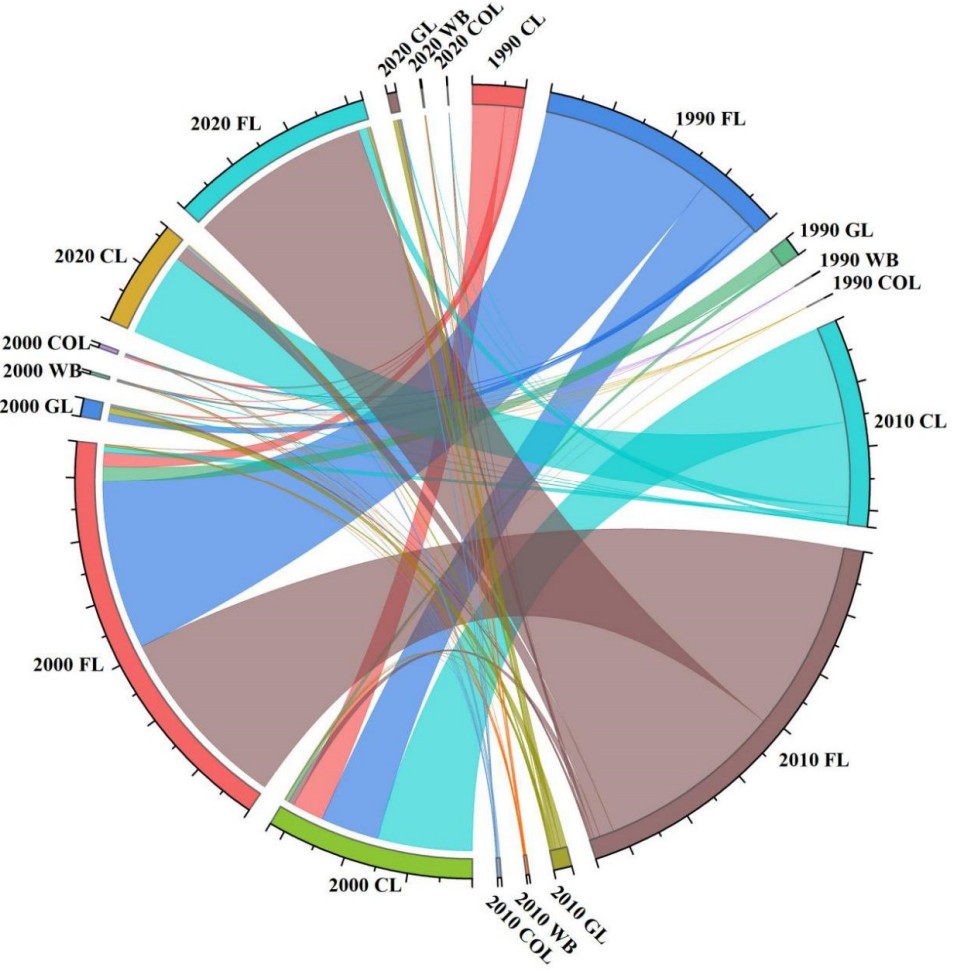

**Figure 4.** Transformation of LUCC during the periods of 1990–2000, 2000–2010, and 2010–2020 in Youyang County. FL: Forestland; GL: Grassland; CL: Cropland, WB: Water body, COL: Construction land.

### 3.2. Dynamics of ESV

Between 1990 and 2000, the ESV rapidly decreased from 6973.29 million yuan to 6283.21 million yuan (Table 3). Among them, the ESV of forestland was the largest reduction of 1087.12 million yuan (Figure 5), and the ESV of cropland increased by 447.29 million yuan. Between 2000 and 2010, the ESV increased from 6283.21 million yuan to 6281.56 million yuan. Between 2010 and 2020, the ESV increased from 6281.56 million yuan to 6410.38, of which the ESV of forestland increased by 113.25 million yuan and that of water bodies increased by 84.76 million yuan.

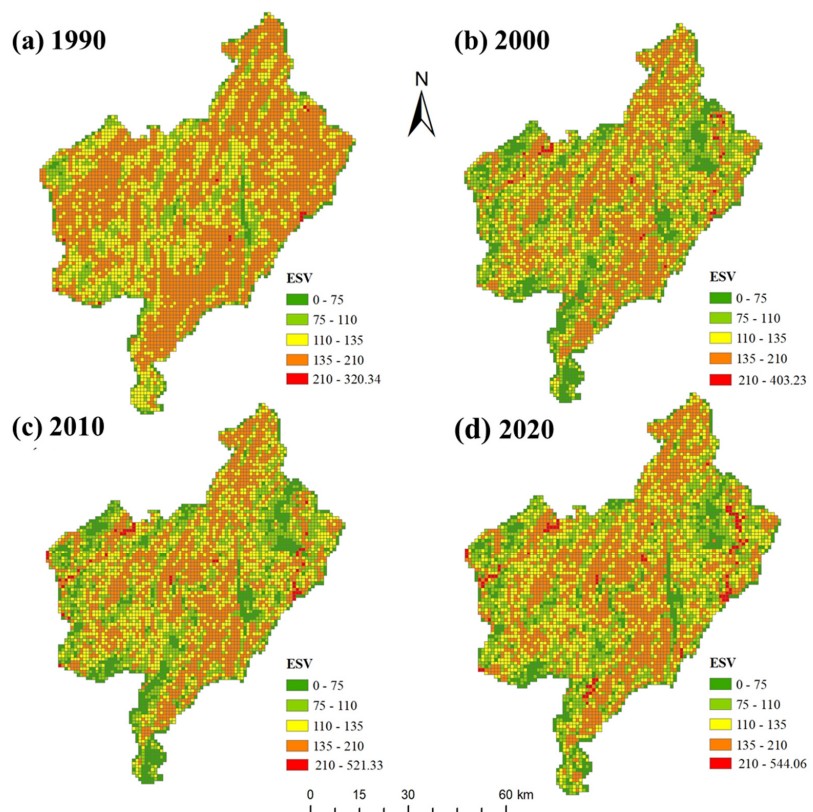

**Figure 5.** Spatial distribution of ESV in 1990 (**a**), 2000 (**b**), 2010 (**c**), and 2020 (**d**).

In terms of service types/functions, climate regulation and water regulation accounted for approximately 50% of total ESV, followed by gas regulation, soil formation and retention, and biodiversity, and the ESV of maintenance of nutrient circulation was the lowest (Table 4). From 1990 to 2020, food production, water supply, water regulation, and maintenance of nutrient circulation increased by 48.12 million yuan, 70 million yuan, 15.17 million yuan, and 38.35 million yuan, respectively. However, the ESV of climate regulation, soil formation and retention, and biodiversity showed the largest reductions of 302.95 million yuan, 106.93 million yuan, and 112.11 million yuan, respectively.

### 3.3. Ecosystem Sensitivity Analysis

The CS of forestland was the highest (0.880999) during the study period (Table 5); that is, when the ESV coefficient increased by 1%, the ESV increased by 0.880999%. The CS of construction land was 0 (Table 5), suggesting that adjusting the VC of construction land does not affect ESV. Overall, the CS of forestland was the highest among all land-use types from 1990 to 2020, while the CS of water bodies gradually increased. Although different land-use types had great differences in different periods, the CSs were 0–0.9 and less than 1, indicating that ESV in Youyang County was not sensitive to the improved VC, and there was an inelastic relationship between them; thus, the research results were credible.

**Table 3.** The ESV of LUCC in Youyang County from 1990 to 2020.

| Year | Cropland ESV (Million Yuan) | % | Forestland ESV (Million Yuan) | % | Grassland ESV (Million Yuan) | % | Water Body ESV (Million Yuan) | % | Construction Land ESV (Million Yuan) | % | Total ESV (Million Yuan) | % |
|---|---|---|---|---|---|---|---|---|---|---|---|---|
| 1990 | 439.16 | 6.30 | 6143.47 | 88.10 | 322.26 | 4.62 | 68.41 | 0.98 | 0.00 | 0.00 | 6973.29 | 100.00 |
| 2000 | 886.45 | 14.11 | 5056.35 | 80.47 | 150.28 | 2.39 | 190.13 | 3.03 | 0.00 | 0.00 | 6283.21 | 100.00 |
| 2010 | 882.43 | 14.05 | 5062.98 | 80.60 | 150.90 | 2.40 | 185.25 | 2.95 | 0.00 | 0.00 | 6281.56 | 100.00 |
| 2020 | 815.10 | 12.72 | 5176.23 | 80.75 | 149.04 | 2.32 | 270.01 | 4.21 | 0.00 | 0.00 | 6410.38 | 100.00 |

**Table 4.** The ESV of ecosystem service functions in Youyang County from 1990 to 2020.

| Ecosystem Service Type | Ecosystem Service Function | 1990 Million Yuan | % | 2000 Million Yuan | % | 2010 Million Yuan | % | 2020 Million Yuan | % |
|---|---|---|---|---|---|---|---|---|---|
| Provision | Food Production | 159.95 | 2.29 | 219.23 | 3.49 | 221.71 | 3.48 | 208.07 | 3.25 |
| | Raw Material Production | 207.30 | 2.97 | 187.53 | 2.98 | 189.66 | 2.99 | 187.54 | 2.93 |
| | Water Supply | 193.28 | 2.77 | 272.09 | 4.33 | 275.16 | 4.31 | 263.28 | 4.11 |
| Regulate | Gas Regulation | 689.75 | 9.89 | 629.27 | 10.02 | 636.40 | 10.02 | 628.33 | 9.80 |
| | Climate Regulation | 1905.02 | 27.32 | 1579.28 | 25.13 | 1597.20 | 25.17 | 1602.07 | 24.99 |
| | Purification of the Environment | 564.49 | 8.10 | 471.44 | 7.50 | 476.79 | 7.51 | 481.17 | 7.51 |
| | Water Regulation | 1398.99 | 20.06 | 1350.14 | 21.49 | 1365.39 | 21.43 | 1414.16 | 22.06 |
| Support | Soil Formation and Retention | 802.02 | 11.50 | 689.69 | 10.98 | 697.51 | 10.99 | 695.09 | 10.84 |
| | Maintenance of Nutrient Circulation | 29.07 | 0.42 | 35.42 | 0.56 | 35.82 | 0.56 | 67.42 | 1.05 |
| | Biodiversity | 710.98 | 10.20 | 589.38 | 9.38 | 596.06 | 9.39 | 598.87 | 9.34 |
| Culture | Recreation, Culture, Tourism | 312.43 | 4.48 | 259.74 | 4.13 | 262.69 | 4.14 | 264.39 | 4.12 |
| | Total | 6973.29 | 100.00 | 6283.21 | 100.00 | 6281.56 | 100.00 | 6410.38 | 100.00 |

**Table 5.** The sensitivity coefficient of Youyang County.

| | 1990 | 2000 | 2010 | 2020 |
|---|---|---|---|---|
| Cropland (VC ± 50%) | 0.062977 | 0.141082 | 0.140479 | 0.127153 |
| Forestland (VC ± 50%) | 0.880999 | 0.804740 | 0.806007 | 0.807476 |
| Grassland (VC ± 50%) | 0.046213 | 0.023918 | 0.024023 | 0.023250 |
| Water body (VC ± 50%) | 0.009810 | 0.030260 | 0.029491 | 0.042121 |
| Construction land (VC ± 50%) | 0 | 0 | 0 | 0 |

*3.4. Driving Mechanism of Spatial-Temporal Heterogeneity in ESV*

3.4.1. Drive Analysis Based on Geo-Detector

The ESV was affected by both natural and human activities factors. The driving factors, except the slope aspect, showed significant correlations with ESV (Table 6). In 1990, the explanation rate of the selected factors for the spatial heterogeneity of ESV was 37.58%. Among the selected factors, soil type and population density had the largest explanation rates (q value) for the spatial heterogeneity of ESV, with 10.34%, and 13.05%, respectively. In 2000, the explanation rate of the selected factors for the spatial heterogeneity of ESV was 34.08%. Therein, altitude, soil type, and population density contributed greatly to the changes in the spatial heterogeneity of ESV, with 8.94%, 7.85%, and 7.44%, respectively. In 2010, the explanation rate of the selected factors for the spatial heterogeneity of ESV was 42.36%. The NDVI, altitude, slope degree, soil type, and population density mostly explained the spatial heterogeneity of ESV, with 8.29%, 9.44%, 5.83%, 7.80%, 6.21%, and 0.06, respectively. In 2020, the explanation rate of the selected factors for the spatial heterogeneity of ESV was 31.36%. The NDVI, altitude, slope degree, and soil type strongly accounted for the spatial heterogeneity of ESV, with 7.36%, 5.52%, 4.97, and 6.02%, respectively. The *q* values ranged from 31.36% to 42.36%, indicating that the selected factors played key roles in the spatial heterogeneity of ESV. It also suggested that more factors were needed to explain the spatial-temporal heterogeneity of ESV in the future.

**Table 6.** The contributions of different driving factors to ESV.

| Year | | NDVI | NPP | Altitude | Slope Degree | Slope Aspect | Lithology | Soil Type | Population Density |
|------|--|------|-----|----------|--------------|--------------|-----------|-----------|--------------------|
| 1990 | q statistic (%) | 1.83 | 1.83 | 1.49 | 5.15 | 3.05 | 0.84 | 10.34 | 13.05 |
| | *p* value | 0.00 | 0.00 | 0.00 | 0.00 | 0.07 | 0.00 | 0.00 | 0.00 |
| 2000 | q statistic (%) | 0.79 | 0.79 | 8.94 | 6.09 | 0.43 | 1.75 | 7.85 | 7.44 |
| | *p* value | 0.69 | 0.69 | 0.00 | 0.00 | 0.27 | 0.00 | 0.00 | 0.00 |
| 2010 | q statistic (%) | 8.29 | 3.08 | 9.44 | 5.83 | 0.27 | 1.44 | 7.80 | 6.21 |
| | *p* value | 0.00 | 0.00 | 0.00 | 0.00 | 0.67 | 0.00 | 0.00 | 0.00 |
| 2020 | q statistic (%) | 7.36 | 2.58 | 5.52 | 4.97 | 0.39 | 0.60 | 6.02 | 4.16 |
| | *p* value | 0.00 | 0.00 | 0.00 | 0.00 | 0.19 | 0.00 | 0.00 | 0.00 |

The interaction effects of any two driving factors were significantly stronger than that of a single factor on the spatial-temporal heterogeneity of ESV (Table 7). Either soil type or population density interacting with other driving factors displayed the highest effect on the spatial-temporal heterogeneity of ESV.

**Table 7.** Interactive effects between the driving factors.

| | NDVI | NPP | Altitude | Slope Degree | Slope Aspect | Lithology | Soil Type | Population Density |
|--|------|-----|----------|--------------|--------------|-----------|-----------|--------------------|
| NDVI | 0.05 | | | | | | | |
| NPP | 0.07 | 0.02 | | | | | | |
| Altitude | 0.14 | 0.11 | 0.06 | | | | | |
| Slope degree | 0.11 | 0.09 | 0.14 | 0.06 | | | | |
| Slope aspect | 0.07 | 0.05 | 0.09 | 0.08 | 0.01 | | | |
| Lithology | 0.08 | 0.05 | 0.08 | 0.07 | 0.03 | 0.01 | | |
| Soil type | 0.16 | 0.14 | 0.19 | 0.15 | 0.11 | 0.12 | 0.08 | |
| Population density | 0.14 | 0.11 | 0.14 | 0.13 | 0.10 | 0.10 | 0.17 | 0.08 |

3.4.2. Drive Analysis Based on Hot- and Cold-Spots

In 1990, the hot-spot regions were mainly distributed in the river valley in the east of Youyang County and the mountainous area of southwest Youyang, while the cold-spot regions were mainly concentrated in the central area of Youyang County and the Longtan

karst trough valley (Figure 6a). In 2000, hot-spot regions and cold-spot regions both increased and expanded. Therein, the hot-spot regions were concentrated in the western river valleys of and in the central area of Youyang County, while the cold-spot regions were mainly concentrated in the area with a low altitude in the south and east of Youyang County (Figure 6b). In 2000, the hot-spot regions and cold-spot regions changed slightly in quantity and spatial distribution (Figure 6c). In 2020, hot-spot regions and cold-spot regions both decreased, except for the cold-spot regions in southern Youyang County. The other spatial patterns of hot- and cold-spots almost did not change (Figure 6d). The spatial differentiation of ESV was driven by many factors. We combined land-use change and selected factors (NDVI, NPP, Altitude, Slope degree, Slope aspect, Lithology, Soil type, and Population density) to visually compare the hot- and cold-spots of ESV change during the study period and qualitatively evaluated the positive or negative moderating effects of policy factors on ESV.

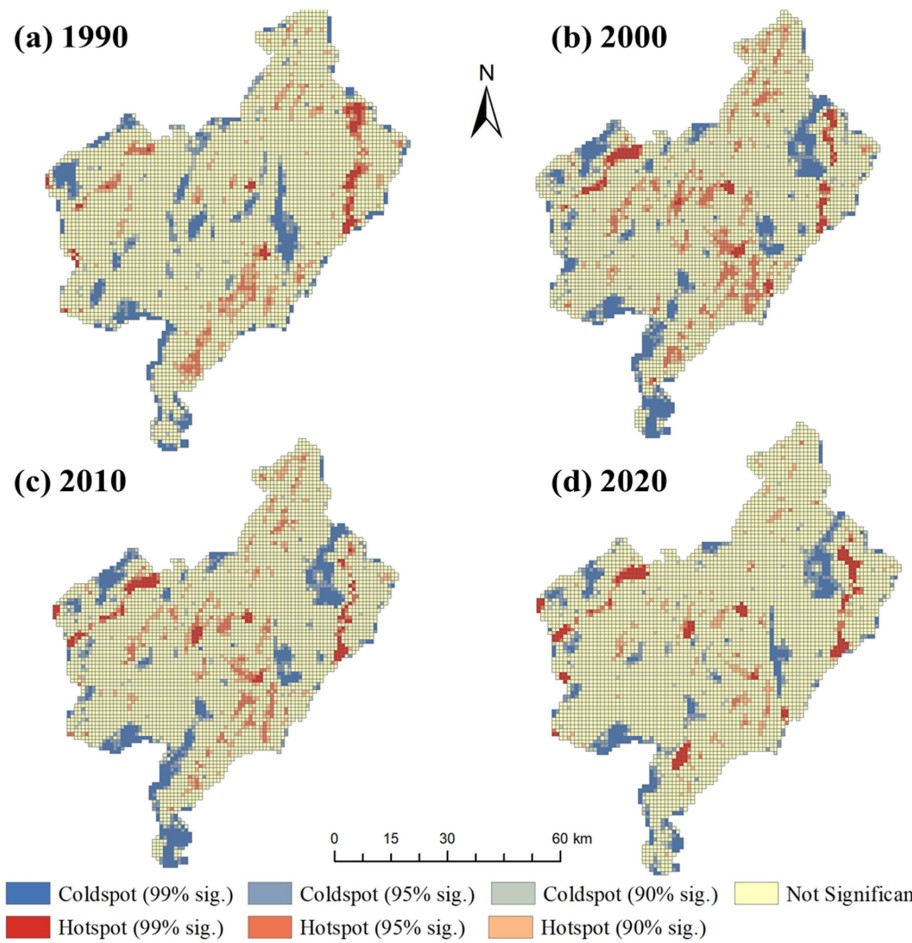

**Figure 6.** Spatial distribution of hot- and cold-spot changes in ESV in 1990 (**a**), 2000 (**b**), 2010 (**c**), and 2020 (**d**). The number in parentheses represents the confidence of the ESV in Youyang County.

## 4. Discussion

### 4.1. ESV Spatial-Temporal Heterogeneity Driven by Natural Factors

Among the seven natural factors, soil type (8.00%) had the strongest effect on the spatial-temporal heterogeneity of ESV, followed by altitude (6.35%) (Table 5). The main geographical background factors affecting the change in ESV were zonal yellow soil and karst lime soil in Youyang County. Soil provides soil ecosystem services like the soil–water cycle, soil biodiversity, and regulation services through soil functions, which in turn affects human well-being and causes the spatial heterogeneity of ESV. The karst area has a shallow soil layer, widely distributed carbonate rocks, and unique hydrogeological processes, lead-

ing to terrain fragmentation and strong spatial heterogeneity of the soil distribution [11,30]. Different soil types have significant differences in soil carbon sequestration and ecosystem service functions [31,32], such as soil formation and retention and the maintenance of nutrient circulation. In addition, soil type affects vegetation type and growth, indirectly resulting in the spatial-temporal heterogeneity of NDVI and NPP. Previous studies have also demonstrated that soil types have significant effects on vegetation types and growth [33,34].

Altitude restricts the distribution of vegetation and the intensity and extent of human activities, thus affecting the spatial heterogeneity of ESV. Wang et al. [35] found that altitude correlated with the total amount of ESV and regulated services value. Hot- and cold-spot analysis also further indicated that the ecological carrying capacity of karst areas was low, and human activities were mainly affected in the middle -low altitude mountain regions. Therefore, the vegetation in high-altitude areas was relatively less affected by human activities and provided higher ecosystem services function, such as Soil Formation and Retention, Climate Regulation, and Purification of the Environment (Figure 6). In contrast, the land-use type of middle-low altitude regions mainly consisted of cropland and construction land, which led to a series of ecological functions like Water Supply, Gas Regulation, Climate Regulation, and Biodiversity reduction. Consequently, the ESV was relatively high. Natural factors, except for soil type and altitude, contributed little (less than 5% for each factor) to the ESV of Youyang County (Table 6). This indicated that there were still some uncertainties in the evaluation of ESV spatial-temporal heterogeneity driven by natural factors, and more factors are required further interpretation in future studies.

Our results also showed that the interaction between soil type and other factor strengthened the effect on the spatial distribution of ESV, indicating that soil type was the most important natural factor restricting the spatial heterogeneity of ESV in Youyang County. The interaction effect of soil type and any other factor on ESV was stronger than that of a single factor (Table 7). Altitude ∩ NDVI and NPP increased the spatial heterogeneity of ESV, suggesting that altitude significantly affected the distribution, type, and growth of vegetation and was one of the important natural factors restricting the spatial heterogeneity of vegetation in Youyang County.

*4.2. ESV Spatial-Temporal Heterogeneity Driven by Human Activities*

LUCC can directly reflect the changes in human activities, and the changes in its pattern and intensity can drive the evolution of ecosystem structure and function, thus driving the changes in ESV [35]. Our results displayed that forestland contributed greatly (accounting for more than 80% of both) to the area and ESV (Table 1, Table 3, and Table 6) from 1990 to 2020, indicating that the basic pattern of ESV was controlled by forestland in Youyang County. This result coincided with the findings of Zhang et al. [14] and Hu et al. [21] in karst areas. In this study, the differences in the area of five land-use types were quite different in the three stages of 1990–2000, 2000–2010, and 2010–2020. Complex dynamic changes occurred among these land-use types in different periods, implying that the ecosystem structure of Youyang County was extremely unstable and poor. This was likely because the ecosystem in the karst area is relatively fragile, and the structure and function of the ecosystem are very sensitive to LUCC changes in this study area [21]. This resulted from the main characteristics of the ecological environment in the karst area [11,30].

Between 1990 and 2000, the main LUCC conversion was from forestland to cropland and resulted in the ESV decreasing by 690.19 billion yuan. The interaction between population density (7.71%) and other driving factors significantly strengthened the driving effect, suggesting that the increase in population density drove the increase in cropland area in Youyang County (Table 7). During this period, farmers reclaimed sloping land at low and middle altitudes, and cropland expanded considerably around construction land, which increased grain yield and raw materials for production and living, resulting in a large decrease in forestland area. As a result, ecosystem service functions such as Climate Regulation, Purification of the Environment, Soil Formation and Retention, and Biodiversity were negatively affected in Youyang County. The results were in good line

with other previous findings [36,37]. Between 2000 and 2010, the DD and land-use transfer matrix were weaker than in the former decade. With the rapid economic growth and gradual urbanization, the construction land around Youyang County increased by 82.01%. Natural ecosystems such as forestland, grassland, and water bodies were replaced by man-made landscapes such as cropland and construction land, reducing the ecosystem services functions such as Water supply and Water regulation. However, because of the area of construction land was relatively small, consequently, the reduction in ESV was also low. Guo et al. [38] also showed that urbanization could lead to a decrease in ESV. Between 2010 and 2020, the urbanization process accelerated, resulting in construction land area increasing by 7 times. However, with the return of sloping cropland and the continuous improvement of water conservancy and irrigation facilities, the forest and water body area increased significantly, thus increasing the ESV.

In addition, population density in Youyang County was very uneven. During the study period, the population density around the administrative center, Longtan karst trough valley, and areas at a low altitude near water sources and cropland gradually increased, leading to a gradual decrease in ESV in these areas. It showed that the concentrated distribution of the populationhad a greater negative impact on the karst ecosystem. The growth in the population inevitably leads to an increase in food demands, which leads to the requisition of a large amount of land for producing food. Han et al. [39] and Wang et al. [40] both showed that the expansion of the population could cause the expansion of food and cropland. In the fragile ecological environment of karst areas, the consequences of ecosystem damage caused by human activities were more serious than those in other areas.

### 4.3. ESV Spatial-Temporal Heterogeneity Driven by Policy Factors

Policy factors showed a very different behavior from other factors, playing a decisive role in influencing ecosystem structure and function, ecosystem trade-offs and synergies, and interacting with other factors [41]. Policy factors, such as Chongqing's Direct Jurisdiction in 1997 and the development of Western China in 1999, promoted the economic development and population growth of Youyang County, expanded cropland, and significantly reduced the forestland and grassland, resulting in the decline in ESV (Figure 7). These policies promoted the change in the spatial distribution pattern and function of land use in Youyang County and increased the area of construction land, thereby affecting the scope and depth of human activity. In addition, population pressure led to conversions among different types of land use (e.g., from grassland and forestland to cropland and construction land). Li et al. [36] and Hou et al. [42] both had similar findings in other areas, showing that urbanization usually promoted the transformation of natural surfaces (cropland and grassland) into artificial surfaces (construction land), which led to a significant decline in ESV.

Since 2000, the policies implemented in Chongqing, such as the Grain-for-Green Program and the Comprehensive Control Project of Rocky Desertification, have converted sloping cropland back to forestland and/or grassland. These policies affected the formation of land-use patterns. These policies improved the ecological structure by not only adjusting the industrial structure but also releasing the rural labor force, thereby affecting the land-use diversity (such as single land-use type in karst areas) and land-use structure [43]. All of these results favored the finding by Zhang et al. [14] who found that land-use change in the karst trough area was affected by the Grain-for-Green Program. The ecological civilization construction initiated in 2012 and the ecological protection and economic development belt in southeast Chongqing, rural revitalization, and other poverty alleviation policies implemented in 2013 had a strong impact on the land-use structure. These policies have accelerated the improvement of the ecological environment and the intensification of urban land use. The ecological restoration policies implemented in the earlier period also began to play a prominent role in this period.

In addition, although this study conducted a qualitative analysis of the policy drivers of ESV, due to the difficulty in quantifying policy factors, it is necessary to find appropriate methods and indicators incorporating into the driver indicator system in the future.

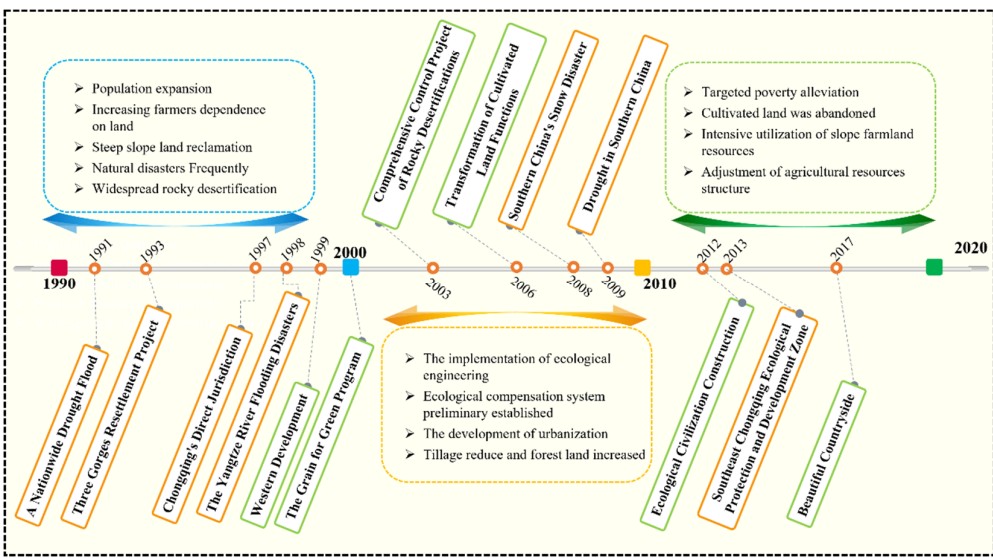

**Figure 7.** The driving mechanism of policies on ESV in karst trough valley.

*4.4. Limitations and Implications of This Study*

Our results indicated that the changes in ESV were driven by many factors, which were not fully quantified in this study. More data are required to study the characteristics of ESV through combined methods in complex conditions such as the karst tough valley here. Indeed, the karst trough valley is one of the important landscapes in the karst ecosystem, and its variations in ESV can partially reflect the ESV characteristics of the whole karst landform. Thus, other karst landforms should be target objectives to fully explore the spatial heterogeneity of ESV in the future. In addition, this study showed the characteristics of ESV evolution in the karst trough valley over the past 30 years, given that the global change and rapid urbanization, the evolution characteristics of ESV can be revealed over a longer duration. We only evaluated the response of ESV to policy changes qualitatively, and the policy factors themselves were not be quantified, which is also an important direction for optimizing ESV evaluations in the future.

In this study, combining natural factors, human activities, and policy factors were effective in elucidating the spatial-temporal heterogeneity of ESV in the typical karst trough valley, SW China. According to the previous studies on ESV in China [21,41], the evaluation of the coefficients of ecosystem services was modified in terms of the ecological environment and social-economic development of Youyang County, and the ESV was estimated on the grid scale (1000 × 1000 m). The spatial-temporal heterogeneity of ESV was directly affected by natural factors and human activities. Meanwhile, policy factors affected the changes in ESV changes through a wide range of driving and/or constraining human activities. The results imply that the direct and indirect effects of nature-human activity-policy on the spatial-temporal variations of ESV should be considered when developing the optimal strategy of ecological restoration in a fragile karst ecosystem.

**5. Conclusions**

This study assessed the spatial-temporal heterogeneity of ESV and its driving factor in the karst trough valley area in southwest China between 1990 and 2020. Forestland was always the dominant land-use type. The dynamic degree of construction land was much higher than other land-use types, attributable to urbanization. The conversion from forestland to cropland dominated the land-use transfer. ESV presented a sharp decrease between 1990 and 2000 and a slow increase between 2000 and 2020. The contribution of

forestland to ESV was more than 80.47%. The occupation of forestland and grassland by cropland resulted in a total reduction in ESV of 562.91 million yuan from 1990 to 2020. Water regulation and climate regulation related to ESV were the highest, followed by gas regulation, soil formation and retention, and biodiversity, and the maintenance of nutrient circulation was the lowest. Soil type explained 8.03% of the spatial-temporal heterogeneity of ESV, followed by population density (7.71%) and altitude (6.34%), indicating that soil type and population density were the most important natural and human activity factors driving ESV change, respectively. Policy factors may affect human activity, thereby altering ecosystem structure and function, ecosystem tradeoffs, and synergy. Our results provide decision support and a more comprehensive scientific basis for the dynamic assessment and driving mechanism of ESV and ecological compensation in karst areas.

**Author Contributions:** Conceptualization, T.L., G.Z., and C.Z.; methodology, C.Z.; software, C.Z., G.Z., and T.L.; validation, B.H.; formal analysis, D.Z. and G.Z.; investigation, T.L. and C.Z.; resources, T.L. and C.Z.; data curation, D.Z.; writing—original draft preparation, C.Z.; writing—review and editing, T.L. and C.Z.; visualization, T.L. and C.Z.; supervision, B.H.; project administration, B.H. and T.L. All authors have read and agreed to the published version of the manuscript.

**Funding:** This study was funded by the National Key Research and Development Program of China (2022YFF1302903), the Innovation Research 2035 Pilot Plan of Southwest University (SWUXDZD22003), the Education and Teaching Reform Research Project of Southwest University (2021JY101), the Fundamental Research Funds for the Central Universities (SWU-KT22060), the Natural Science Foundation of Chongqing, China (CSTB2022NSCQ-MSX0385), and the State Cultivation Base of Eco-agriculture for Southwest Mountainous Land, Southwest University.

**Data Availability Statement:** The original contributions presented in the study are included in the article, further inquiries can be directed to the corresponding author.

**Conflicts of Interest:** The authors declare no conflicts of interest.

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
