# Peer review of "Spatial-Temporal Heterogeneity of Ecosystem Service Value Driven by Nature-Human Activity-Policy in a Representative Fragile Karst Trough Valley, SW China"

_land, doi:10.3390/land13020256_

Round 1
Reviewer 1 Report
Comments and Suggestions for Authors
In general, I found this a useful paper which describes land use changes in an ecologically sensitive part of China over a period of several decades. My main reservations had to do with some of the methodology (which I fully admit I am not completely familiar with), and the interpretation and explanation of the results. In short, I would like to see more explanation for uninformed readers and I make suggestions about this below.
At the heart of the paper is the estimation of ecosystem service values (ESVs). It appears from line 153 that estimated monetary values for ESVs are drawn from a paper by Costanza et al and another by Xie et al. I cannot find any discussion in the paper about how relevant these values are for this
case study. I think it is important to discuss this, as many of those values were developed in other countries and other situations. I found section 2.4 a little difficult to follow. It seems that a monetary value for grain production was not available from the sources mentioned above and had to be derived from other sources. It was not clear to me why 1/7th of the economic value of grain production in the current year was chosen (although reference 21 was referred to). As an aside, I found it interesting that construction land was given 0 value, when some discussions of ecosystem services would ascribe some ecosystem-related aspects to construction land (for example, dwellings
built and sold on the basis of the environment they are located in and/or the views and amenities that they provide are sometimes considered to have ecosystem service values). I understand why the authors have ascribed a 0 value, and this is not a major criticism from my point of view.
More importantly however I think that there needs to be more, and simpler, explanation in the discussion about why, and how, ecosystem service values change as the mix of land uses changes. Currently, the discussion talks about associated changes in land use, especially the amount of forested area, and reports the resulting changes in ecosystem service values without going into much explanation for the reader about how this relates to the different functions performed by different land uses and how the value of these functions in relation to human demands and needs is assessed. I suggest that for a general readership this sort of explanation would be very useful. For example, there is a lot of forest in this study, which provides a range of ecosystem functions and services. As a short-cut to value those services, a value from the literature has been used. However, the reader should be made aware that real value must be considered in relation to supply and demand. If, for example, there is more forested than required to meet human demand, then a small loss of forested land might not be very important and the need to conserve forested land might not be as great as if the supply was below or only slightly above demand. Similarly, if demand for food is critical, then conversion of some forested land to cropland might be seen as economically and socially desirable. I know that these considerations are difficult to take into account in a study like this, but I suggest that some recognition should be given in the Discussion.
Mostly, the English expression is good but in places it could do with a bit of editing. I give some examples below. But, more generally, there is a tendency to use overly emotive, and sometimes unsubstantiated, language. For example:
• lines 33 to 34: “irrational land use”. No evidence is given to support this claim and, anyway, it is not necessary. Replacing “irrational land use” with “land use” would be enough
• lines 51 to 52: “unreasonable human activities”. The same applies as in the dot point above.
Below I give specific comments about different parts of the manuscript, which I hope, if addressed, will improve it.
Lines 41 to 42: “they used sensitivity analysis to regulate the influence of the coefficient on the estimation of ESV”. Unclear sentence.
Lines 55 to 56: “given the severe harsh of karst ecosystem”. This could be rewritten as “given the harshness of karst ecosystems”.
Section 2.2, Data Sources: I am not familiar with the analytical methods used in this study, so I am not able to comment on the adequacy of the data sources or the methods used. They seem sound to me.
Section 2.6.2, Hot and Cold Spots Analysis: I found this section difficult to understand. It is not clear to me what Gi is. From the second sentence (lines 190 to 192) it seems that hot spots are areas of high ESV and cold spots are areas of low ESV. However, the third sentence refers to analysing the response of spatial patterns of ESV to policy changes and it is not clear to me how this was done. I suggest rewriting this section so that it is much clearer to an uninformed reader.
Tables are very clear and informative, as are the figures. Figures 2 and 4 four are particularly interesting and summarize a lot of information very clearly. The results are reported succinctly and explained clearly.
Section 3.4.1: I would like to see a little more discussion about these percentages. Although these driving factors have a significant relationship with ESV, they really only explained a small proportion and the variation. I think this needs to be explained to the reader.
Section 3.4.2: I am not necessarily critical of this analysis, but I note that the definition of hot and cold spots depends largely on the monetary values given to different ecosystem services. As discussed above, these values are derived from other circumstances and places. As the authors will realize, value is determined by both supply and demand, and it would be worth discussing the effect of factors such as population (i.e., amount of demand) and lifestyle (i.e. type of demand) and how these might affect estimates of ecosystem service values. I recognise that some of this is discussed in
the Discussion (for example the effect of population growth on land use change), but I think we need to be careful about interpreting labels like hot and cold spots without considering the reliability of the monetary values attached to different ecosystem services. These factors will be important for
decision makers when applying policies for land use.
Lines 321 to 322: Again, I draw attention to the low percentage of the spatial temporal heterogeneity of ESV that is explained by soil type and altitude, as discussed above. I think we need to be careful to not make too big a deal of this correlation.
Lines 332 to 337: Again, I urge caution in this type of interpretation. The fact that altitude is correlated with total amount of ESV is largely because forests grow at higher altitudes and the ESV itself is determined by the estimates of monetary value attached to the ESVs, so we need to be careful about conclusions like “economic development and ecological protection should be
enhanced in the areas of low elevation and ecological protection should be strictly performed in the areas with high elevation”. I do not necessarily disagree with these conclusions, but we need to be careful when basing them simply on ESVs. For example, there are other ecological reasons why
these recommendations could be made (e.g., arguments to do with the mix of ecosystem services, the relative rarity of some services and the extent of the services in relation to demand).
Section 4.2: The statements made in this section are true, but they tend to be more like reporting of results than interpretation. For example, it is true that forested land dominated the area and that the main conversion of land was from forest to cropland. I suggest that there is a need for more interpretation of these results. Yes, conversion of forest to cropland might be seen as undesirable if only the monetary values used in this study and total changes in area of different land uses are considered. However, policymakers will need to consider other factors not taken into consideration, such as the number of people who rely on services from forests versus other land use types, the types of demand these people have, the number of lives enhanced by the availability of crops and the implications of that for health and well-being. My point is that basing policies simply on ecosystem service values is risky and there is a need to consider broader social issues as well. The authors begin to touch on some of these aspects around line 402, where it is recognized that growth in population is a strong driver of demand for food and, therefore, conversion of land.
I note that in lines 407 to 410 that there is reference to “excessive human activity” that is said to deteriorate habitat quality etc. The judgment that human activity is excessive is subjective and not something that is explained quantitatively in this study. I suggest that it is enough to note that some
forms of human activity can change the mix of land uses which can affect biodiversity and therefore the types of ecosystem services that are available. It must be remembered that the ability to produce food is itself an ecosystem service, the value of which needs to be considered in relation to social demands and values.
Section 4.3: This is a useful discussion, which explores how various policies that have been implemented during the study might have affected changes in land use and therefore ecosystem services. I urge caution in the way conclusions are drawn here. For example, it is reasonable to note that these policies probably encouraged certain types of land use, but to conclude that they were responsible for the changes in a scientific study would have required some sort of consideration of what changes might have happened without the policies (i.e., a control) or if alternative policies had been put in place. In other words, while it is reasonable to suggest that these policies have had an effect, this study was not set up to provide proof that these policies resulted in the land use changes.
Figure 7: In relation to my points above, I think this is a good diagram that suggests how policies may have driven land use change. I think this is a useful set of ideas to discuss, but, as discussed above, it is speculation.
In my comments above, I have mostly concentrated on constructive criticisms with only a few positive observations about the many things that are good about this paper. I hope this does not give the authors the impression that I am not impressed with this piece of work. I think it is very useful. My main suggestion is that some of the implications be explained to readers in ways that give them a greater understanding of the importance of ecosystem services in relation to human needs and goes beyond simple reporting of results. This should not require a major rewrite. As well as a few sentences inserted in places I have suggested, the section on limitations (section 4.4) could highlight issues like the complexity of estimating value fully and the difficulty of attributing outcomes to particular policies.
The quality of English language is high with some small issues to be addressed as discussed in my report
Reviewer 2 Report
Comments and Suggestions for Authors
The paper aimed to investigate the spatio-temporal heterogeneity of the ESV during the period 1990-2020 in a karst valley located in southwestern China through the analysis of soil evolution to which a geo-detector statistical analysis (sensitivity analysis) and analysis of hot and cold points was then applied to understand the interactions between the ESV and natural factors, human activity and the factors that drive policy. The results showed that from 1990 to 2020, forest land accounted for more than 62.32% of the total area, and building land increased continuously due to urbanization, and ESV fell sharply in 1990-2000 and slowly increased from 2000 to 2020, resulting in a total reduction of 562.91 million yuan in ESV, due in large part to the occupation of forests and pastures by cultivated land.
The topic is certainly interesting but, in my opinion, the work has some weaknesses that do not allow it to be published in its current form:
o Firstly, the evolutionary dynamics of land cover are obvious because they are classic dynamics that have affected all rural areas that have undergone processes of development of agriculture and urbanization.
o the authors should better clarify why karst areas are environmentally critical.
o A critical aspect is the Table 1. The equivalent coefficient of ESV because it is not clear how these values were determined and/or from which source they were taken. The monetary valuation of ecosystem services is one of the most critical elements that must be addressed in this type of analysis and there is a large literature dealing with this topic ... No reference is made here to how these parametric values were determined.
o Among other things, I would like to ask whether, over time (about 30 years), these services have always had the same value, or has it changed? How do researchers deal with this problem?
o Another critical aspect is the choice of the analysis grid (1000x1000 m = 100 ha) which is too large and which, inevitably, leads to make approximate evaluations of the changes because I think it is difficult to find homogeneous pixels for land use. In this case, how were the pixels that contained a mix of land uses (e.g. arable part, part forest, part urbanized) considered?
o To say that soil type is the main variable that explains the spatio-temporal heterogeneity of ESV is a contradiction in terms because this is, in my opinion, the variable being evaluated and, therefore, how can we say that it is, at the same time, cause and effect of change?
o A final consideration on which I believe the authors could take a cue to deepen the analysis are the effects of policies which, in statistical elaborations, are defined in a general way but which, at the same time, are listed in a precise way (see also Fig. 7). These policies, among other things, have pursued quite opposite objectives over time (e.g. increasing production in a first phase, restoring forests and grasslands in a second phase, etc.). I believe that an interesting starting point for analysis could be precisely that of verifying the qualitative and quantitative effects of these policies on land use in order to verify their ex-post effectiveness.
Round 2
Reviewer 2 Report
Comments and Suggestions for Authors
The paper has improved compared to the first version and the authors have responded adequately to the criticisms I made in the first revision.
The authors have, in fact, improved the critical points both by inserting some explanatory paragraphs and by inserting new and important bibliographical references.
From my point of view, the paper is now publishable.